# The Role of Organ and Leaf Habit on the Secondary Xylem Anatomy Variation across 15 Species from Brazilian Cerrado

**Rafaella Dutra** [1,2,*], **Anselmo Nogueira** [3], **Sergio Rossi** [1], **Larissa Chacon Dória** [4], **Valentina Buttò** [1,5] and **Carmen Regina Marcati** [2]

1   Laboratoire sur les Écosystèmes Terrestres Boréaux, Département des Sciences Fondamentales, Université du Québec à Chicoutimi, 555 Boulevard de l'Université, Chicoutimi, QC G7H 2B1, Canada
2   Laboratório de Anatomia da Madeira, Departamento de Ciência Florestal, Solos e Ambiente, Faculdade de Ciências Agronômicas, Universidade Estadual Paulista (UNESP), Botucatu 18610-034, SP, Brazil
3   Centro de Ciências Naturais e Humanas, Universidade Federal do ABC, São Bernardo do Campo 09210-580, SP, Brazil
4   Departamento de Botânica, Instituto de Biologia, Universidade Estadual de Campinas (UNICAMP), Campinas 13038-970, SP, Brazil
5   Forest Research Institute, Université du Québec en Abitibi-Témiscamingue, Rouyn-Noranda, QC J9X 5E4, Canada
*   Correspondence: remdutra@etu.uqac.ca

**Abstract:** Xylem is a complex tissue connecting the organs of plants and it performs multiple functions, including water transport, mechanical support, and storage. Because of the interaction between structure and function, xylem anatomy can provide useful information about its role in plant strategies. However, knowledge of how xylem anatomical traits change across organs and species functional groups is still limited. Here, we tested the role of different plant organs (stem and roots) and leaf habits (deciduous, semi-deciduous, and evergreen) on xylem anatomy variation across 15 woody species from the Brazilian Cerrado. Vessels, fibers, and parenchyma traits were measured on 45 individuals sampled in 2014 in Botucatu, São Paulo, Brazil. Our results revealed a higher parenchyma fraction and less fiber fraction in roots than in stems across species. Differences in wood anatomical traits between organs were mainly species-specific in parenchyma traits rather than vessel and fiber traits. Across leaf habits, only the root ray fraction was higher in evergreen species compared to deciduous species. These findings highlight a potential role of organs and leaf habits in xylem storage across Cerrado woody species.

**Keywords:** functional traits; wood allocation space; xylem plasticity; ray parenchyma; neotropical savanna

## 1. Introduction

The emergence of secondary xylem (i.e., wood) is a milestone in ensuring the dominance and longevity of plants in terrestrial ecosystems [1]. As part of the vascular system connecting the components of the plant body, xylem plays multiple functions, such as long-distance water transport, mechanical support, and storage. For most angiosperms, these functions are performed by different cell types: vessels conduct water, fibers provide mechanical support, and the parenchyma (rays and axial) stores nutrients and water. Considering such complexity, dividing the xylem area amongst different cell types could result in functional trade-offs, enhancing some functions over others [2,3]. Thus, the enhancement of certain xylem functions due to xylem cell partition allows plants to maximize fitness and survival [4]. Wood anatomical traits, therefore, provide a detailed insight into tissue properties and functionality as well as ecological and adaptive species strategies [5]. However, critical questions about how wood anatomy changes across organs and species functional groups (e.g., leaf habit) still awaits a clear answer.

Explanations of the wood functionality based on anatomy are well made at the stem and branch level [6–10], while variation in belowground plant traits remains poorly quantified. However, considering that the xylem is a functional unit that integrates plant compartments, inferring wood traits from a single or aerial organs may obscure the implications of wood functionality at the whole-plant level [5]. Therefore, how different plants' functional requirements above- and belowground reflect on xylem anatomy remains challenging [11,12]. Xylem traits' variability is expected to mirror the demands for efficient water transport, storage, and mechanical support of roots and stems. For example, wider conduits indicate greater hydraulic conductivity [13] and are often observed in roots given the need for efficient water transport from the soil to leaves [14,15]. A higher amount of parenchyma tissue also implies storage demand and less investment in support cells in roots due to the soil matrix surrounding underground organs [16–18]. By contrast, the xylem profile tendency in the stem, considering the bottom–top direction, shows thicker cell walls and narrower cell lumen diameter to deal with the expected needs for mechanical support (i.e., canopy support and wind load breaking resistance) [11,19] or water transport failure resistance under drought-induced tension [20,21], or both requirements simultaneously.

Under different conditions, species can have leaf habits responding to seasonal environmental filters, with varying traits linked to water and carbon use and acquisition strategies [22,23]. Thus, leaf permanence or fall are potentially connected to xylem anatomy [24,25]. Plants can be classified into functional groups according to leaf habit senescence, as deciduous (i.e., shedding leaves at the peak of the dry season), intermediate such as semi-deciduous plants (i.e., partial shedding in the dry season), and evergreen (i.e., retaining leaves throughout the year) [26]. Compared to evergreens, deciduous species exhibit a larger vessel diameter [24], enhancing efficient water transport and photosynthetic leaf capacity [27,28]. However, larger vessel diameters imply greater vulnerability to hydraulic failure via embolism [29]. Therefore, evergreen species tend to show a hydraulic system that better avoids embolism than deciduous species. In addition, due to the higher demand for non-structural carbohydrate allocation during the regrowth period [30], higher parenchyma fractions are expected for carbohydrate storage in species with a deciduousness strategy. Although studies between deciduous and evergreen indicate differences in xylem anatomical traits linked to water transport [7,23,24,31,32], how leaf habit explains xylem anatomy variation beyond these classifications and in the view of storage function is not well understood.

Here, we investigated the role of plant organs (i.e., stem and roots) and leaf habits (i.e., deciduous, semi-deciduous and evergreen) on wood anatomy variation in a functional perspective. For this, we collected woody samples from 15 species in the Brazilian Cerrado (Neotropical savanna), where plants' growth and survival are driven by water seasonality, fire dynamics and soil fertility [33]. We hypothesized that (1) there is a higher investment in water transport efficiency (i.e., larger vessels) and storage (i.e., higher parenchyma amount), and less investment in support (i.e., fiber fraction and wall thickness) in roots compared to the stem; and (2) evergreen species have embolism-resistant water transport (i.e., narrow vessels, higher vessel density and thicker fibers) and less investment in storage (i.e., parenchyma amount) than deciduousness species strategy.

## 2. Materials and Methods

### 2.1. Study site and sampling

The study was conducted in Estancia Santa Catarina Private Reserve, Botucatu (São Paulo State, Brazil) (22°54′51″ S, 48°30′13″ W). The site is located in the Cerrado sensu stricto (i.e., Brazilian savanna), with vegetation composed of short and sparse woody species [34]. The soils are sandy and acid, with low organic matter and high aluminum content (Departamento de Solos e Recursos Ambientais, UNESP Botucatu, São Paulo, Brazil). The area has a mean annual temperature of 21 °C and a seasonal precipitation pattern of 1507 mm from September to April and 50 mm from May to August.

We selected 15 dominant species based on a previous floristic inventory in the area and covering a wide phylogenetic diversity (Figure 1, Table 1). In addition, all species have diffuse-porous wood except *Aegiphila verticillata* Vell., with semi-porous rings. We sampled the stem and root from three mature individuals per species, at 60 cm aboveground and a depth of 15–30 cm from the root collar. Samples were collected in June–July 2015, when the vascular cambium is dormant [35].

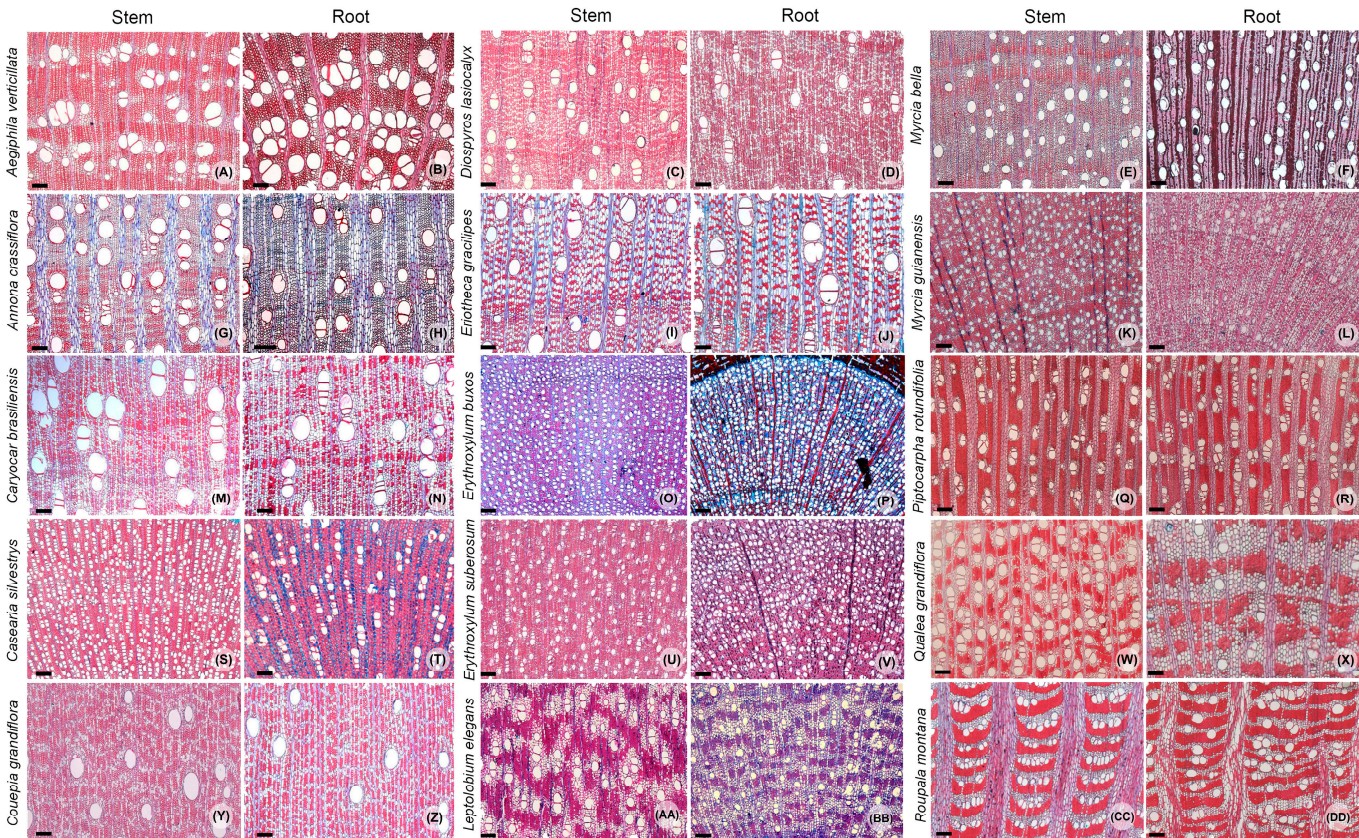

**Figure 1.** Stem and root wood, in the transversal section, of studied Cerrado Brazilian plants. For leaf habit type, see Table 1. Scale bars= 200 μm.

**Table 1.** Characteristics of studied species. Values represent means ± standard error (N = 3). Stem and root diameters were measured at 60 cm aboveground and 15–30 cm belowground, respectively. Dec: deciduous; Sd: semi-deciduous; Ev: evergreen.

| Species | Family | Leaf Habit | Growth Form | Plant Height (m) | Stem Diameter (cm) | Root Diameter (cm) |
|---|---|---|---|---|---|---|
| *Aegiphila verticillata* Vell. | Lamiaceae | Dec | Tree | 4.3 ± 0.7 | 14.7 ± 3.9 | 3.6 ± 1.3 |
| *Annona crassiflora* Mart. | Annonaceae | Dec | Tree | 3.8 ± 0.6 | 15.7 ± 2.3 | 8.8 ± 0.7 |
| *Caryocar brasiliense* Cambess. | Caryocaraceae | Sd | Shrub | 1.8 ± 0.1 | 5.7 ± 2.3 | 12.4 ± 2.0 |
| *Casearia sylvestris* Sw. | Salicaceae | Sd | Shrub | 2.3 ± 0.1 | 4.7 ± 2.0 | 1.5 ± 0.2 |
| *Couepia grandiflora* (Mart. & Zucc.) Benth. | Chrysobalanaceae | Dec | Tree | 3.1 ± 0.5 | 12.5 ± 1.0 | 11.9 ± 0.4 |
| *Diospyros lasiocalyx* (Mart.) B. Walln. | Ebanaceae | Dec | Tree | 2.7 ± 0.2 | 9.2 ± 0.5 | 7.5 ± 0.5 |
| *Eriotheca gracilipes* (K.Schum.) A.Robyns | Malvaceae | Sd | Tree | 4.6 ± 0.8 | 19.3 ± 2.4 | 13.6 ± 0.9 |
| *Erythroxylum buxos* Peyr. | Erythroxylaceae | Ev | Shrub | 2.2 ± 0.4 | 2.7 ± 0.3 | 2.2 ± 0.2 |
| *Erythroxylum suberosum* A.St.-Hil. | Erythroxylaceae | Sd | Tree | 2.7 ± 0.6 | 9.7 ± 2.2 | 2.7 ± 1.4 |
| *Leptolobium elegans* Vogel | Leguminosae | Dec | Tree | 4.5 ± 1.0 | 13.3 ± 1.8 | 2.8 ± 0.6 |
| *Myrcia bella* Cambess. | Myrtaceae | Sd | Tree | 4.5 ± 0.5 | 11.9 ± 0.5 | 5.7 ± 2.5 |
| *Myrcia guianensis* (Aubl.) DC. | Myrtaceae | Ev | Shrub | 2.1 ± 0.1 | 2.9 ± 0.2 | 1.8 ± 0.3 |
| *Piptocarpha rotundifolia* (Less.) Baker | Asteraceae | Ev | Tree | 3 ± 0.3 | 14.6 ± 3.5 | 7.5 ± 3.2 |
| *Qualea grandiflora* Mart. | Vochysiaceae | Dec | Tree | 5.8 ± 0.7 | 26.7 ± 4.7 | 13.5 ± 9.2 |
| *Roupala montana* Aubl. | Proteaceae | Ev | Tree | 3.0 ± 0.5 | 6.3 ± 1.5 | 6.6 ± 2.6 |

*2.2. Wood Anatomy Measurements*

Samples were cut in transversal and tangential sections 15–22 μm thick with a sliding microtome (Reichert Om E, Vienna, Austria). Sections were stained in safranin and astra blue, dehydrated in ethanol, and mounted on permanent slides with synthetic resin (Entellan®, Merck, Darmstadt, Germany). To measure individual cellular elements, samples were reduced in small fragments and placed with hydrogen peroxide and glacial acetic acid (1:1) in an oven at 60 °C for 24 h. The resulting material were stained with safranin and mounted on semi-permanent slides with 50% glycerin. Measurements were performed using light microscopy (Axioscop 40 Zeiss, Jena, Germany) equipped with Axiocam MRc camera (Zeiss, Jena, Germany) and Axiovision software (Version 4.9.1, White Plains, NY, USA). Anatomical wood traits were measured with ImageJ 2.0 [36]. All measurements were performed on one cross-section, except for ray density that was measured on tangential sections, and fiber features from individualized elements for each sample [37,38]. In total, we analyzed 1335 vessels (660 in roots, and 675 in the stem), 1200 fibers (630 in roots, and 570 in the stem) for fiber lumen diameter, and 1260 fibers (630 in both organs) for fiber wall thickness. The xylem traits of each sample were averaged to determine individual values. Vessel and ray densities (number/area) were estimated in 1 mm$^2$ and 1 mm$^{-1}$, respectively, across 10 fields. Fiber lumen diameter and wall thickness (μm) were calculated on 30 individual fibers. Hydraulic vessel diameter (Dh) was calculated based on the formula $(\Sigma d^4/n)^{1/4}$, where *d* refers to vessel diameter, estimated from vessel lumen area, and *n* refers to vessel number [13]. The vessel lumen area was measured in 30 vessels per individual. The vessels, fibers, and parenchyma fractions were quantified over an area of 1 mm$^2$ [39].

*2.3. Data Analysis*

To test for differences between organs, we used linear mixed models (LMM) on anatomical wood traits (response variables) as a function of organs (root; stem) (fixed effects) across species, in which the individuals nested in species were considered as random term in the models. For each species separately, we also compared anatomical wood traits between organs using Student's *t*-test, the Welch *t*-test or the Mann–Witney U-test according to data normality checked a priori using the Shapiro–Wilk test and histograms. Plant height and organ diameter size were also included as fixed factors due to their potential allometric effect on the anatomical wood traits [40–42].

To test for differences between leaf habits, we used LMM on anatomical wood traits (response variables) as a function of leaf habits (deciduous; semi-deciduous; evergreen) (fixed factor) for each organ separately. Species was considered as a random term in the models. A Bonferroni post hoc test was applied for multiple comparisons among leaf habits, respectively.

All mixed models were compared by the likelihood ratio test. The normality and homoscedasticity of the residuals were checked by visual inspection [43]. When needed, data were transformed by square root or logarithm to meet the normality or homogeneity of variances. Analyses were performed in R (Version 4.0.5, Vienna, Austria) [44]. The lme4 package [45] was used to fit LMM, and the lmerTest package [46] and emmeans package [47] were used to test the fixed effects significance of models and for multiple comparisons, respectively.

## 3. Results

Wood anatomical traits differed slightly between root and stem across species (Figure 2, Table 2). On average, the hydraulic vessel diameter was similar between organs (root: 76.32 ± 6.25 μm vs stem: 70.74 ± 5.40 μm) (Figure 2A), with highest values observed in roots of *Diospyros lasiocalyx* and *Eriotheca gracilipes*, about 5.5-fold higher than the lowest values observed in both organs of *Erythroxylum buxos* (Figure 3A). When contrasted organs within species, *Leptolobium elegans* showed larger hydraulic vessel diameter in the stem (root: 47.48 ± 6.34 μm vs stem: 76.55 ± 5.89 μm; t = −3.36, df = 4, *p* = 0.03, Figure 3A). Vessel

density showed the largest variability across organs, ranging between 1.70 and 193.63 n° mm$^2$ in roots and between 3 and 164.70 n° mm$^2$ in the stem; no difference, on average, was observed between organs (Table 2). Within species, we detected a higher density of vessels in stems *in Couepia grandiflora* (t = −3.15, df = 4, $p$ = 0.03) and *Roupala montana* (t= −3.14, df = 4, $p$ = 0.04) (Figure 3C). Fiber and parenchyma properties were less than about 1% divergent between roots and stems across species (Figure 2E,G,I). On another hand, both cell types features differed between organs at intraspecific level. Wider fiber lumen diameters were detected in the roots of *C. grandiflora* and stems of *Myrcia guianensis* (t = 3.48, df = 4, $p$ = 0.02; t = −2.82, df = 4, $p$ = 0.05, respectively) (Figure 3E). For fiber wall thickness, the higher values were found in stems of *Annona crassiflora* (t = −19.92, df = 2, $p$ = 0.002) and in roots of *Casearia sylvestris* (t = 4.87, df = 4, $p$ = 0.008) (Figure 3G). Compared to the stem, ray density was ca. 16% lower in the roots of *C. sylvestris* (t= −4.88, df = 4, $p$ = 0.008, Figure 3I).

**Table 2.** Fixed effect estimates of linear mixed models for organ (root, stem), organ diameter and plant height on wood anatomical traits across 15 Cerrado species. Individuals nested within species were considered the random term in all models. Values in bold denote statistical significance. SE = standard deviation.

| Response Variable | Fixed Factor | Estimate | SE | t-Value | *p*-Value | Graph |
|---|---|---|---|---|---|---|
| sqrt Hydraulic vessel diameter | Organ (root–stem) | 0.16 | 0.33 | 0.47 | 0.64 | Figure 2A |
| | Organ_diameter | 0.01 | 0.04 | 0.20 | 0.84 | |
| | Plant_height | 0.38 | 0.24 | 1.61 | 0.11 | |
| Vessel fraction | Organ (root–stem) | 0.00 | 0.00 | 0.37 | 0.72 | Figure 2B |
| | Organ_diameter | 0.00 | 0.00 | 0.51 | 0.62 | |
| | Plant_height | 0.00 | 0.00 | 1.69 | 0.10 | |
| log Vessel density | Organ (root–stem) | 0.10 | 0.12 | 0.75 | 0.46 | Figure 2C |
| | Organ_diameter | **−0.03** | **0.01** | **−2.15** | **0.04** | |
| | Plant_height | 0.03 | 0.09 | 0.32 | 0.75 | |
| Fiber fraction | Organ (root–stem) | **−0.07** | **0.03** | **−2.54** | **0.02** | Figure 2D |
| | Organ_diameter | −0.00 | 0.01 | −0.13 | 0.90 | |
| | Plant_height | 0.01 | 0.02 | 0.30 | 0.77 | |
| Fiber lumen diameter | Organ (root–stem) | −0.04 | 0.37 | −0.09 | 0.93 | Figure 2E |
| | Organ_diameter | 0.01 | 0.04 | 0.10 | 0.92 | |
| | Plant_height | −0.07 | 0.26 | −0.26 | 0.80 | |
| Total parenchyma fraction | Organ (root–stem) | **0.07** | **0.03** | **2.27** | **0.05** | Figure 2F |
| | Organ_diameter | 0.01 | 0.01 | 0.11 | 0.78 | |
| | Plant_height | −0.02 | 0.02 | −1.05 | 0.48 | |
| Fiber wall thickness | Organ (root–stem) | −0.33 | 0.21 | −1.57 | 0.07 | Figure 2G |
| | Organ_diameter | 0.02 | 0.23 | 0.51 | 0.55 | |
| | Plant_height | 0.06 | 0.13 | 0.46 | 0.17 | |
| Ray fraction | Organ (root–stem) | **0.08** | **0.02** | **3.79** | **0.002** | Figure 2H |
| | Organ_diameter | **0.01** | **0.01** | **2.75** | **0.04** | |
| | Plant_height | −0.02 | 0.02 | −1.53 | 0.06 | |
| sqrt Ray density | Organ (root–stem) | 0.02 | 0.09 | 0.18 | 0.97 | Figure 2I |
| | Organ_diameter | **0.02** | **0.01** | **2.19** | **0.05** | |
| | Plant_height | −0.10 | 0.06 | −1.90 | 0.77 | |
| Axial parenchyma fraction | Organ (root–stem) | 0.01 | 0.02 | 0.43 | 0.72 | Figure 2J |
| | Organ_diameter | −0.01 | 0.02 | −0.15 | 0.76 | |
| | Plant_height | −0.01 | 0.01 | −0.95 | 0.79 | |

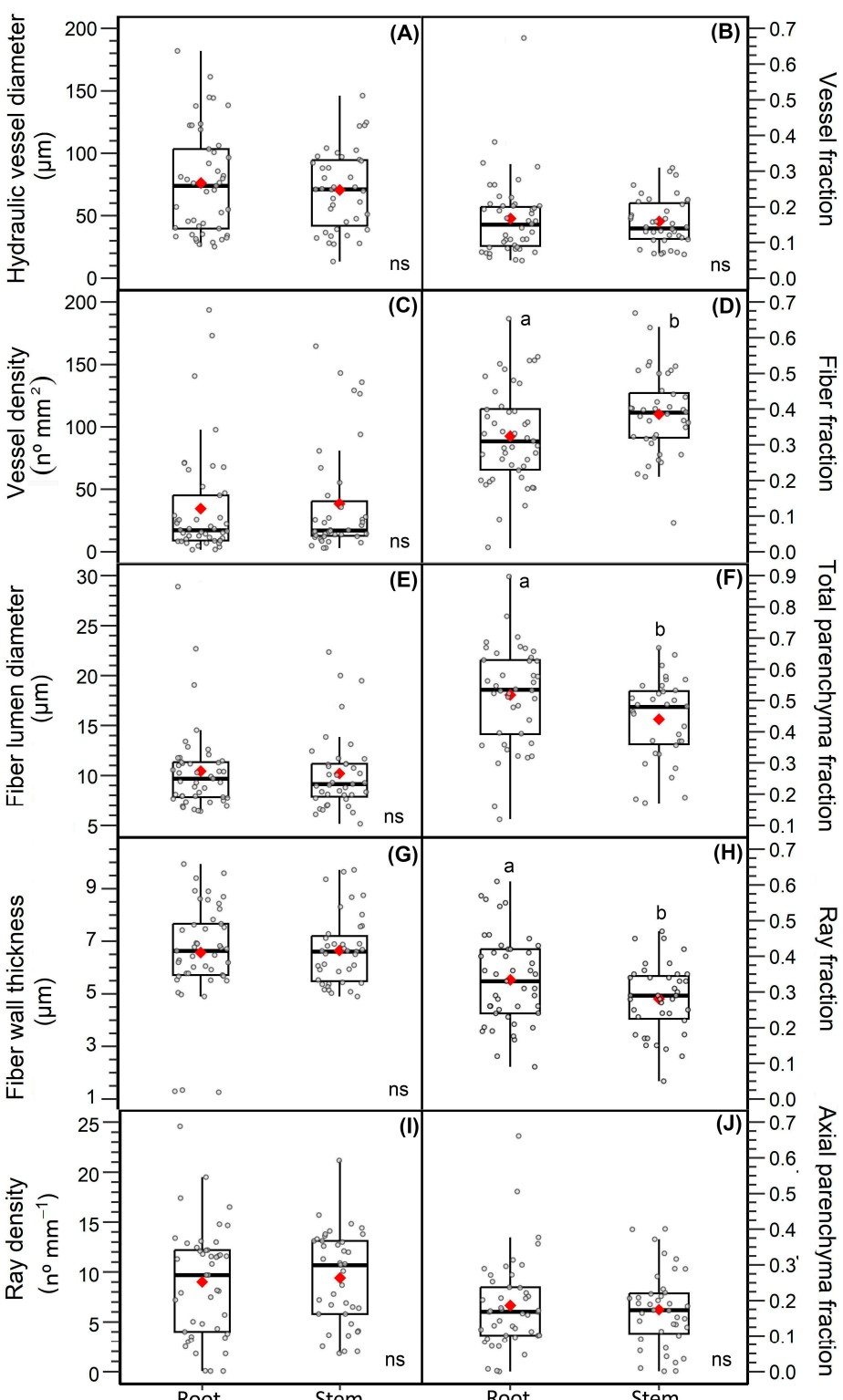

**Figure 2.** Anatomical xylem traits (i.e., hydraulic vessel diameter (**A**); vessel fraction (**B**); vessel density (**C**); fiber fraction (**D**); fiber lumen diameter (**E**); total parenchyma fraction (**F**); fiber wall thickness (**G**); ray fraction (**H**); ray density (**I**); axial parenchyma fraction (**J**)) as function of organs across 15 woody species from Cerrado. Boxplots show the average (red diamond), individual data points (gray dots), and median (horizontal line); lower and upper boxplots represent the 25th and 75th percentiles; and whiskers represent 1.5 × the interquartile range. Different letters indicate significant difference between organs according to linear mixed models (*p* < 0.05). ns = not significant.

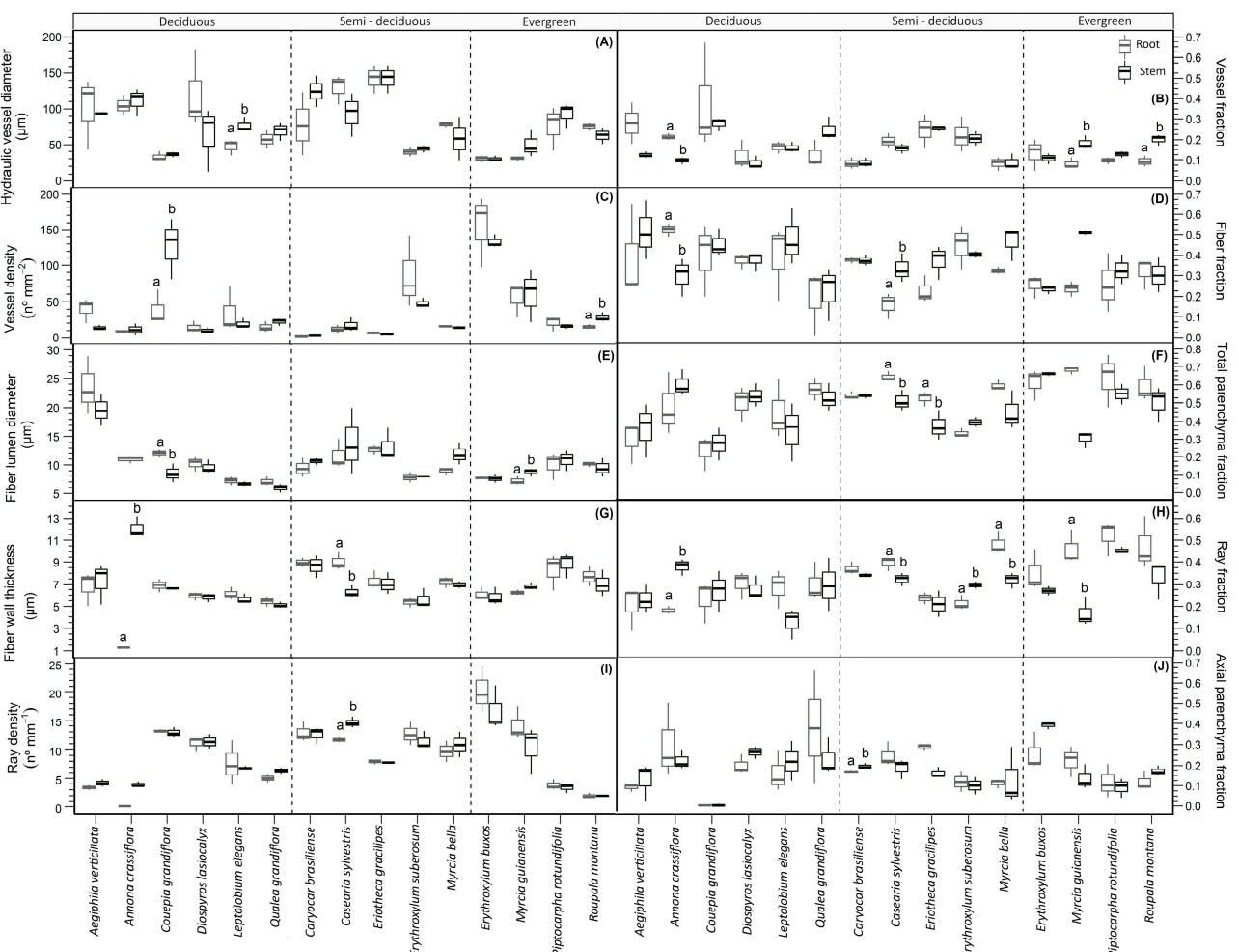

**Figure 3.** Anatomical xylem traits (i.e., hydraulic vessel diameter (**A**); vessel fraction (**B**); vessel density (**C**); fiber fraction (**D**); fiber lumen diameter (**E**); total parenchyma fraction (**F**); fiber wall thickness (**G**); ray fraction (**H**); ray density (**I**); axial parenchyma fraction (**J**)) as function of species, organs, and leaf habits from Cerrado. Roots and stems are shown in gray and black, respectively. Boxplots show the median (horizontal line); the lower and upper boxplot represent the 25th and 75th percentiles, and whiskers represent 1.5 × the interquartile. Different letters indicate significant difference between organs for a given species ($p < 0.05$).

At the cell fraction level, both organs were composed mainly of total parenchyma fraction (root: 0.51; stem: 0.46), which varied about 7.5-fold in roots and 4-fold in stems (Figure 2F). The total parenchyma fraction was driven by the ray fraction (Figure 2H), and both fractions were larger in roots compared to stems across the studied species ($p = 0.05$; $p = 0.0002$, Table 2). *C. sylvestris* and *Eriotheca gracilipes* showed higher total parenchyma fraction in the roots than stems (Figure 3F). Ray fraction showed different patters between organs, with lower values in roots of *A. crassiflora* and *Erythroxylum suberosum* (t = −8.56, df = 4, $p = 0.001$; t = −3.36, df = 3, $p = 0.05$, respectively) and higher values in roots of *C. sylvestris*, *Myrcia bella* and *M. guianensis* (t = 2.86, df = 4, $p = 0.05$; t = 5.02, df = 4, $p = 0.009$; t = 5.02, df = 4, $p = 0.007$, respectively) (Figure 3H). In contrast, the fiber fraction was slightly greater in stems (0.38 ± 0.01) than in roots (0.32 ± 0.01) ($p = 0.02$, Table 2). Roots of *A. crassiflora* and stems of *C. sylvestris* showed higher fiber amounts (t = −3.18, df = 4, $p = 0.04$; t = 4, df = 4, $p = 0.02$, Figure 3D). Across species, no difference was detected in the vessel and axial parenchyma fractions in roots relative to stems (Table 2). The amount of vessels in the roots was higher in *A. crassiflora* and lower in *M. guianensis* and *R. montana* (t = 9.35, df = 4, $p <0.0001$; t = -4.38, df = 4, $p = 0.02$; t = -5.04, df = 4, $p = 0.007$, Figure 3B). Likewise, lower root axial parenchyma fraction was

found in *Caryocar brasiliense* (root: $0.17 \pm 0.003$ vs stem: $0.20 \pm 0.009$; t = $-2.98$, df = 4, $p = 0.04$, Figure 3J).

Among leaf habits, a higher similarity was detected in wood anatomy for each organ (Figure 4, Table S1). From the 10 anatomical wood traits measured, only the ray fraction changed among leaf habits (F = 9.08, $p = 0.003$), with roots of evergreen species showing higher values than deciduous species ($p = 0.04$) (Figure 4H). Intermediate values were found in semi-deciduous species (Table S1). Across species, roots of *Piptocarpha rotundifolia*, *R. montana* and *M. bella* showed the highest ray amount, and the stem of *L. elegans* showed the lower (Figure 3H).

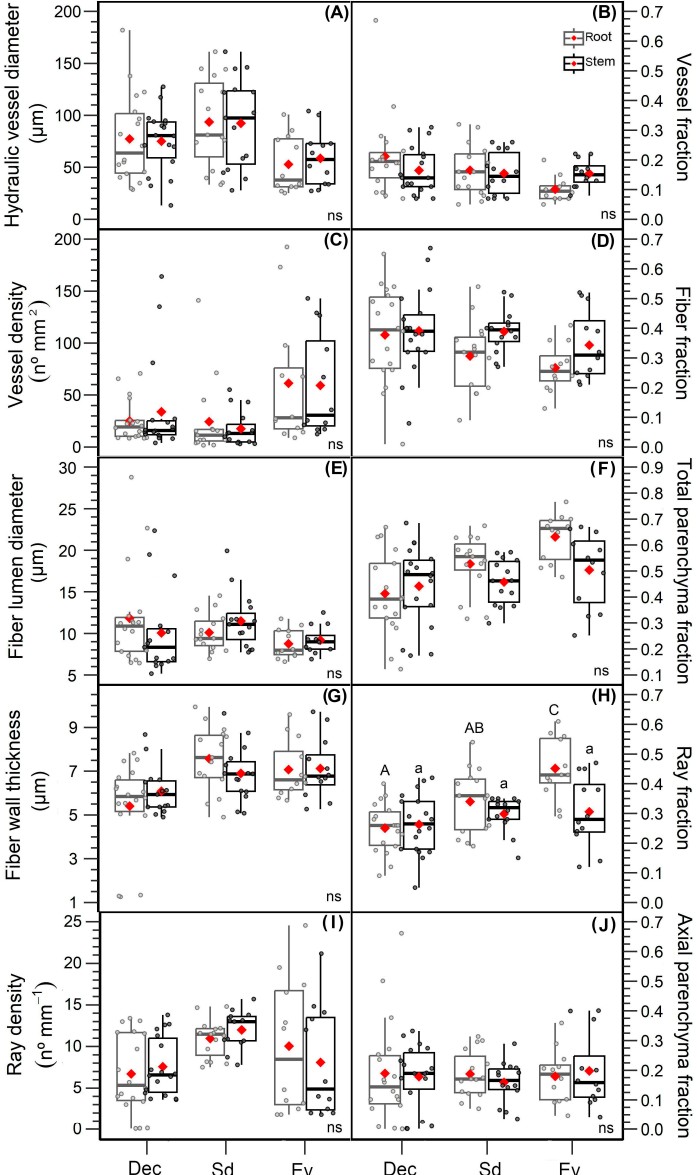

**Figure 4.** Anatomical xylem traits (i.e., hydraulic vessel diameter (**A**); vessel fraction (**B**); vessel density (**C**); fiber fraction (**D**); fiber lumen diameter (**E**); total parenchyma fraction (**F**); fiber wall thickness (**G**); ray fraction (**H**); ray density (**I**); axial parenchyma fraction (**J**)) as function of leaf habit senescence and organs of the studied Cerrado species (N = 15). Roots and stems are shown in gray and black, respectively. Boxplots show the average (red diamond), individual data points (dots), and median (horizontal line); lower and upper boxplots represent the 25th and 75th percentiles; and whiskers represent 1.5× the interquartile range. Different letters indicate a significant difference among leaf habits for a given organ according to Bonferroni post hoc test ($p < 0.05$); uppercase: root, lowercase: stem. ns = not significant. Dec, deciduous; Sd, semi-deciduous; Ev, evergreen.

We also examined the potential influence of organ diameter and plant height on the anatomical traits. Vessel and ray densities as well as ray fraction were correlated with organ diameter (slope = −0.03, *p* = 0.04; slope = 0.02, *p* = 0.05; slope = 0.01, *p* = 0.04, respectively) (Table 2), but the relatively low slope estimates indicate no influence of organ diameter on either trait. No studied trait correlated with plant height (Table 2).

## 4. Discussion

Evaluating xylem anatomical traits variation between organs and functional plant groups is crucial to map the functional meaning of the complex xylem tissue, determining plant fitness and forest function. In this study, we identified shifts of anatomical xylem traits comparing different organs and leaf habits categories across 15 species from Cerrado. Overall, our results indicate that: (1) organs are related to storage and, to a lesser extent, support tissue variation; (2) contrasts in wood anatomical traits between organs are mainly species-specific in parenchyma traits than vessel and fiber traits; (3) at the organ level, divergences in root parenchyma amount are best explained by leaf habits. Collectively, these findings indicate that organ and leaf habit have a major role in wood storage across the species of Cerrado plants studied.

### 4.1. Xylem Anatomy Variation between Organs

Contrasting functional demands and environments of plant organs are expected to translate into xylem traits and functions. The higher amount of parenchyma (total and ray) and lower amount of fiber in roots than in stems across species confirmed our hypothesis that storage rather than mechanical support is emphasized in roots compared to stems. In agreement with previous studies [12,17,18], a greater amount of parenchyma indicates higher storage capacity in roots. In comparison, a greater amount of fiber in the stem indicates a requirement for mechanical support [19]. This pattern possibly reflect distinct biomechanical demands within the plant body and the microenvironment that surrounds the organs. Indeed, the soil matrix supports anchoring the plants, reducing the requirement for mechanical support tissues and favoring storage for roots compared to aerial organs such as stems [16]. Thus, our findings demonstrate that xylem parenchyma and fiber fractions variation are structured by organs differently, which underlie potential trade-offs between storage and support functions.

Similarly, these patterns above were also detected in some species studied when comparing organs within the same species. For example, roots of *C. sylvestris* and *E. grandiflora* showed a higher total fraction of parenchyma, and *M. bella*, *M. guianensis* and *C. sylvestris* showed a higher ray fraction, while the stem of *C. sylvestris* showed a higher fiber fraction (Figure 3F,H,D, respectively). However, lower root ray fractions were observed in *A. crassiflora* and *E. suberosum*, indicating that in both species, ray parenchyma may be a limited proxy of storage function in the root wood. Furthermore, *A. crassiflora* exhibited a higher root amount of vessels and fibers. This implies a potential way to minimize the wood risks of embolism induced by water stress [21,29].

The findings that organs differentiate within species could explain the large overlap of most anatomical xylem features between organs across species. Indeed, certain anatomical wood traits in below- and aboveground organs are variable at the intraspecific level [17,48]. For example, we found a larger stem vessel diameter in *L. elegans* and a higher stem vessel density and amount in *C. grandiflora* and *R. montana*, and *M. guianensis*, respectively. These results indicate that xylem in root and stem is structured to avoid failures in water transport via embolism [29], and it is potentially species-specific for these species. However, implications concerning wood anatomy and water transport should also consider the pit membrane features due to effects on water transport over long distances [14,49], and they need to be further evaluated across Cerrado species. Likewise, observed divergences in fiber features between root and stem in a given species suggest that fiber function as a proxy of wood functions is organ and species dependent. Among the species studied, the large fiber lumen diameter in roots of *C. grandiflora* and stems of *M. guianensis* indicates a dual

fiber function of support and storage [50,51]. On the other hand, the thicker fiber observed in stems of *A. crassfilora* and roots of *C. sylvestris* potentially reinforces certain contrasting demand for mechanical support at the intraspecific level. Finally, other species-specific characteristics (e.g., ray cellular dimensions [52]) could explain the higher ray density in the stem of *C. sylvestris*.

Despite the patterns discussed above, most of the studied species showed convergences in wood anatomical traits in roots compared to stems. There are, however, other possible explanations. First, general xylem features can be more variable according to environmental, taxonomic classifications (e.g., genus, family), life history or genetically conserved [51,53], which may minimize the organ or species-specific influence on wood traits. Second, patterns of phenological organ growth dynamics vary substantially within species, as observed in roots of temperate tree species [54,55]. Further studies which take these variables into account are needed to better understand structural–functional trade-offs in xylem and plant demands above- and belowground in a broader perspective.

In addition, we observed that no studied trait correlates with organ diameter or plant height across Cerrado species. This outcome is contrary to previous studies which have demonstrated that anatomical wood characteristics variation is influenced by plant size [40–42,56]. One potential explanation is that the effect of plant or organ size on variations in anatomical traits could be obscured when considering a single sampling position. Indeed, a potential plant size allometric effect on the anatomical wood traits was detected when considering different heights from the base to tip of the stem in wide range of species [40,50] or along the root and stem [41,42]. Moreover, plant size-dependent wood anatomy variation can occur in species-specific anatomical traits [56], and it is different from those investigated in this study. Future studies along the root and/or stem across and within species from Cerrado is therefore recommended.

### 4.2. Xylem Anatomy Variation among Leaf Habits

When considering the association of contrasting leaf habits and xylem anatomy, we only found a higher root ray fraction in evergreen plants. This result is contrary to our expectation of increased storage investment in species with deciduous species. The space allocated to ray parenchyma allows for greater storage capacity in woody tissue for the studied evergreen plants [57]. Conversely, less parenchyma in deciduous plants implies a limited role of wood parenchyma as a carbohydrate storage component. The requirement for more energy reserves for regrowth [30] leads deciduous species to prioritize storage tissues and cell types. We suggest that the sampled root deciduous species show alternative carbohydrate storage strategies, such as living fibers and inner bark [58–60]. Notwithstanding, our results diverge from the pattern reported of xylem anatomical traits variability related to water transport and support functions among species with contrasting leaf habits [7,23,24,31,32]. Therefore, regardless of organ, the leaf habit potentially plays a limited role in variation in the vessel, fiber, and axial parenchyma characteristics across woody plants from the Brazilian savanna.

### 5. Conclusions

Our study provides a more integrated insight into the wood anatomy and functional traits of woody plants, reinforcing the importance of plant organs, leaf phenological groups and species-specific analyses to deepen knowledge on xylem variation and plant functionality. The counterbalancing of parenchyma and fiber fractions between roots and stems indicates potential contrasts in mechanical demands on different plant compartments that reflect a trade-off between storage and mechanical support functions. Contrasts in xylem parenchyma traits between organs vary mainly at the intraspecific level. By comparing evergreen, semi-deciduous and deciduous species, we provide evidence that leaf habit is a suitable predictor for variation in storage capacity in root xylem. These findings show that organs, intrinsic factors (i.e., species-specifics), and leaf habits play a major role in structuring wood anatomical variation in parenchyma traits across species from Cerrado. Our approach, based on functional groups represented by leaf habit, in the most floris-

tically diverse savanna [61], also provides valuable information to detect and generalize plant growth response patterns and better understand their responses under extreme climatic scenarios.

**Supplementary Materials:** The following is available online at https://www.mdpi.com/article/10.3390/f14020269/s1. The Supplementary Material describes wood anatomy traits as function of the leaf habit of species studied. Table S1. Fixed effect estimates of linear mixed models for leaf habits (Dec: deciduous, Sd: semi-deciduous, Ev: evergreen) on wood anatomical traits for a given organ (root, stem) across 15 Cerrado species. Species were considered as a random factor in the models. A Bonferroni post hoc test was used for multiple comparisons among leaf habits. Values in bold indicate statistical significance ($p < 0.05$).

**Author Contributions:** Conceptualization and investigation: R.D. and C.R.M.; methodology: C.R.M., R.D. and A.N.; formal analysis and writing—original draft preparation: R.D.; writing—review and editing R.D., S.R., L.C.D., V.B., A.N. and C.R.M.; supervision: C.R.M. and A.N.; funding acquisition: C.R.M. R.D. was corresponding author. All authors have read and agreed to the published version of the manuscript.

**Funding:** This study was supported by the Fundação de Amparo à Pesquisa do Estado de São Paulo (FAPESP—Procs. 15/14954-1, 19/09417-8), and the Consejo Nacional de Ciencia y Tecnologia (CONACYT—Proc. 237061). R.D. received grants from the Coordenação de Aperfeiçoamento de Pessoal de Nível Superior (CAPES), Emerging Leaders in the Americas Program (ELAP—Global Affairs Canada), the Observatoire régional de recherche en forêt boréale, and Centre d'étude de la forêt (CEF—Fonds de recherche du Québec). C.R.M. thanks the Brazilian National Council for Scientific and Technological Development for the productivity grant (CNPq—Proc. 304715/2018-2), S.R. thanks CAPES-PrInt/Sustainable Development for the grant (Finance Code 462), and A.N. thanks the Brazilian National Council for Scientific and Technological Development for the Research Grant (CNPq—Proc. 434692/2018-2) and São Paulo Research Foundation through a Young Investigators Grant (FAPESP—2019/19544-7).

**Data Availability Statement:** Data are available upon request to the corresponding author.

**Acknowledgments:** We are grateful to O. Pupo for access to the Cerrado reserve. We thank P. Vergilio, J. A. Rosell, M. Olson, F. Bosio, M. Blagitz, L. C. Pereira, J. da Silva, T. Oya and J. M. Santos for the fieldwork, L.C. Pereira for technical assistance in the lab, and A. Garside for checking the English text. This study is part of the master's dissertation of the first author.

**Conflicts of Interest:** The authors declare no conflict of interest.

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
