# Peer review of "The Role of Organ and Leaf Habit on the Secondary Xylem Anatomy Variation across 15 Species from Brazilian Cerrado"

_forests, doi:10.3390/f14020269_

Round 1
Reviewer 1 Report
The paper "The role of organ and leaf habit on the secondary xylem anatomy variation across 15 species from Brazil Cerrado" written by Dutra et al. is worth publishing, contains a lot of interesting data, but the manuscript as it is, needs to be significantly improved to be published in Forests.
The main doubts relate to the methods of data collection, the measurements methods and further interpretation. In my opinion, combining data, collected from trees of different species, although it undoubtedly gives a general picture, should be preceded by a detailed analysis of each species separately. Additionally, the results are also too cursorily discussed.
Main concerns:
Material and methods
1. Why ring-porous wood was not analyzed - an analysis of a different species, with a diverse wood structure could give many interesting results. I understand, of course, that dominant species were taken into analysis and now it would be difficult to perform additional analyses for ring-porous wood. But the Authors should add some information and own interpretation considering e.g. if in their opinion diffuse-porous trees are better suited for Brazilian savanna and are they dominant due to that feature?
2. What do the authors mean by longitudinal section - tangential or radial? On what microtome were the preparations cut - please specify the brand and model; How many measurements were taken and how many sections were analyzed from each tree? Were data from three trees of one species averaged? How many cells were measured? Do all cells measure per 1mm2? On what sections (transverse?) were the measurements made? What were the longitudinal sections used for? How old were the tested trees?
Results
3. Grouping trees of all species together and comparing the anatomical features of stem and roots wood, raises the greatest doubts. It is known that the anatomy of the secondary xylem of these two organs differs in many species. Therefore, the study lacked a comparative analysis of the root and stem wood structure of individual species. Only after this analysis it would be possible to draw conclusions and present more general results. The Authors only refer, and for selected xylem features only, to individual species, indicating which of the features had the lowest or highest values, but these are only minor annotations. Therefore, in my opinion, the Fig S1 should be introduced as one of the main results of the work, because it shows the differences between stem and root for specific species, which does not always correlate with the general trend for the entire group. The difference between specific species and the general trend of the group were not discussed at work, and should be added.
4. Fiber fraction and total parenchyma fraction are greater in stems or roots, respectively (lines 167-170), but this is not the case if we look at the individual species in Fig S1. The Authors do not refer to these data in the result section nor discuss them. What can cause such discrepancies in different tree species?
5. The authors did not refer to the form of plants (trees vs. shrubs). It is known that trees and shrubs differ in wood structure. Comparative statistical analysis should be done for species with different forms. Very interesting results would probably be obtained from such a study. For example, lines 147-149 – vessel density showed the largest variability across organs… - this may be the result of differences between individual species related to the form of growth (trees, shrubs).
6. In addition, I have the greatest doubts about some of the analysis of the anatomical features of wood in relation to leaf habits (lines 177-196). As I understand, the Authors compared the wood traits obtained from the measurements of stems and roots in relation to leaf habits. The results grouped in this way may not be reliable and the conclusions may not be correct. In my opinion, the Authors should compare the anatomical structure of wood from stems and from roots in relation to leaf habits separately. In this part of the paper, the Authors refer to Fig S1 where the results obtained for roots and stems are shown separately - so it is somewhat misleading. Which data was analysed - root and stem separately or together.
7. Lines 198-202 – why the Authors do not observe any correlations related to plant height or organ diameter. Such correlations in individual species are described in the literature. Please refer to this in the discussion.
8. The discussion is written in a too general way, as I mentioned in earlier comments.
Minor comments:
Line 150 - Table 2 – add Fig 2c and S1c
Line 152 – A. grandiflora – A. verticillate or C. grandiflora
Reviewer 2 Report
This study tests the role of different plant organs (stem and roots) and leaf habits (deciduous, semi-deciduous, and evergreen) on xylem anatomy variation across 15 woody species from the Brazilian Cerrado. The article is well-written and reads fluent. The figures and the tables are clear. These findings highlight a potential trade-off between xylem functions underlying the anatomical differences between organ and leaf habits across the Cerrado woody species. However, before considering it for publication, the following comments need to be addressed:
1. Line 25-26, “evergreen species had a higher fraction of parenchyma than semi-deciduous”, but see from figure 3, this should be revised as evergreen species had a higher fraction of ray parenchyma than semi-deciduous.
2. Line 44-45, “However, critical questions about how wood anatomical changes are influenced across organs and species are still not well understood”. But there are many literatures about these questions, please add them.
3. Line 103-104, “at 60 cm aboveground and a depth of 15 – 30 cm belowground”, What is the basis of your choice?
4. Table 1, please add the stand age of all the species.
5. Line 156, there was a return key.
6. In table 2 and 3, the unit of vessel density was μm?
7. Line 196-197 “The vessel, total parenchyma and axial parenchyma fraction values were not contrasting among leaf habit types”, total parenchyma fraction should be deleted here.
8. Line 259, “a lower density of vessels is a way to minimize the risks of embolism induced by water stress”, is this correct?
Round 2
Reviewer 1 Report
The authors addressed all suggestions and comments appropriately and made the necessary changes throughout the manuscript.
I received a text for review in which the changes were marked in red. This was very confusing because the authors did not mark all the changes they made.
I ask the authors to check if there is no mistake in lines 170 - 172:
For fiber wall thickness, the higher values were found in the roots ( or stems?) of Annona crassiflora (t = -19.92, df = 2, P = 0.002), and in the stem (or roots?) of Casearia sylvestris (t = 4.87, df = 4, P = 0.008) (Figure 3G).
And if so, please check the whole text again in details.